# VizXP: A Visualization Framework for Conveying Explanations to Users in Model Reconciliation Problems [*]

**Ashwin Kumar, Stylianos Loukas Vasileiou, Melanie Bancilhon, Alvitta Ottley, William Yeoh**

Washington University in St. Louis

ashwinkumar@wustl.edu, v.stylianos@wustl.edu, mbancilhon@wustl.edu, alvitta@wustl.edu, wyeoh@wustl.edu

## Abstract

Advancements in explanation generation for automated planning algorithms have moved us a step closer towards realizing the full potential of human-AI collaboration in real-world planning applications. Within this context, a framework called *model reconciliation* has gained a lot of traction, mostly due to its deep connection with a popular theory in human psychology, known as the *theory of mind*. Existing literature in this setting, however, has mostly been constrained to algorithmic contributions for generating explanations. To the best of our knowledge, there has been very little work on how to effectively convey such explanations to human users, a critical component in human-AI collaboration systems. In this paper, we set out to explore to what extent visualizations are an effective candidate for conveying explanations in a way that can be easily understood. Particularly, by drawing inspiration from work done in visualization systems for classical planning, we propose a visualization framework for visualizing explanations generated from model reconciliation algorithms. We demonstrate the efficacy of our proposed system in a comprehensive user study, where we compare our framework against a text-based baseline for two types of explanations – domain-based and problem-based explanations. Results from the user study show that users, on average, understood explanations better when they are conveyed via our visualization system compared to when they are conveyed via a text-based baseline.

## Introduction

From its inception, *Explainable AI Planning* (XAIP) has garnered increasing interest due to its role in designing explainable systems that bridge the gap between theoretical and algorithmic planning literature and real-world applications. The primary motivation of XAIP systems has been centered around creating well integrated pipelines that, given different personas of human users (the explainees),[1] they can generate explanations of a plan for a given planning problem. One of the recurring themes in this context is the *model reconciliation problem* (MRP) (Chakraborti et al. 2017) – a seminal work that utilizes a popular theory in human psychology, called the *theory of mind*,[2] and allows an agent (the explainer) to consider the "mental model" of the user[3] in its explanation generation process. These explanations bring the model of the user closer to the agent's model by transferring a minimum number of updates from the agent's model to the user's model. However, most of the effort on this topic has mostly focused on algorithmic contributions for generating explanations. To the best of our knowledge, there has been very little work on how to effectively communicate and convey the explanations generated to users. For instance, the current state-of-the-art by Sreedharan et al. (2020) presents explanations as text, typically in the PDDL format, which can, arguably, increase the user's misunderstanding of the task, especially for novice users.

A well-established educational principle, called the *multimedia learning principle*, posits that humans learn better from words and pictures, than from words alone (Mayer 1997). For example, Clark and Mayer (2016) showed that accompanying text-based instructions with pictures improved students' performance on a test by a median amount of $89\%$. Interestingly, students got around $65\%$ of answers correct after seeing a combination of text and pictures, compared to less than $40\%$ of answers correct after reading a text comprised of words alone. Similar results have also been obtained in object assembly tasks (Brunyé, Taylor, and Rapp 2008). As such, there is strong evidence within the psychology community that the use of visual content has a profound effect on increasing retention and comprehension when compared to text alone.

Based on this principle, in this paper, we set out to explore to what extent visualizations constitute an effective candidate for conveying explanations (in an MRP setting) in a way that can be easily understood by human users. In particular, by drawing inspiration from work done in visualizing classical planning problems, we propose a visualization framework that can visualize the action-space and state-space of

---

[*]This research is partially supported by National Science Foundation grants #1812619, #1838364, and #2020289.

[1]The current norm in the XAIP literature considers the following three personas: *end user*, *domain designer*, and *algorithm designer* (Chakraborti, Sreedharan, and Kambhampati 2020).

[2]The theory of mind is the ability to attribute mental states (beliefs, intents, knowledge, etc.) to others and recognize that these mental states may differ from one's own.

[3]The mental model is just the user's version of the problem which the agent possess, and interestingly, it can be expressed as a graph, a PDDL model, or even a logic program.

planning problems, and use it as a medium for communicating explanations between an agent and a user. In addition, we introduce two taxonomies of explanations that can be visualized by our framework: (1) *Domain-based explanations*, which arise due to discrepancies between the action models of the agent and the user, and (2) *Problem-based explanations*, which arise due to differences in the initial or goals states of the agent and the user. Our proposed framework is agnostic to how explanations are generated and is thus *orthogonal* to all algorithmic contributions for model reconciliation problems. In summary, we make the following contributions: (1) We propose a visualization system for visualizing explanations in MRP settings; (2) We define two types of explanations – domain-based and problem-based explanations; (3) We demonstrate the efficacy of our proposed system in a comprehensive user study, where we compare our framework against a text-based baseline. Results from the user study show that users, on average, understood explanations better when they are conveyed via our visualization system compared to when they are conveyed via a text-based baseline.

## Related Work

The fundamental problem we are addressing in this paper is formulated around the *model reconciliation problem* (MRP) (Chakraborti et al. 2017) within the XAIP literature. In an MRP, the plan of a planning agent is unacceptable to a human user due to differences in their models of the problem. As such, the agent needs to provide an explanation of that plan in terms of model differences. In this context, researchers have tackled MRP from various perspectives, such as traditional search-based methods (Sreedharan et al. 2020), MDP-based models and approaches (Sreedharan et al. 2019), and logic-based formulations (Vasileiou, Previti, and Yeoh 2021).[4] Nonetheless, as we mentioned in the introduction, existing work has mostly focused on developing algorithms for generating explanations, and not on how they are to be conveyed to a human user; a common thread is that the explanations are communicated to users through text messages.

There has also been some effort by the planning and scheduling community to create user interfaces for planning and scheduling problems (Freedman et al. 2018). While some work aims to show users the space of alternate plans (Gopalakrishnan and Kambhampati 2018; Magnaguagno et al. 2020; Chakraborti et al. 2018), others aim to create systems to aid users in the creation of plans (e.g., Planimation (Chen et al. 2020)) or for assistance with domain modeling (e.g., Conductor (Bryce et al. 2017)).[5] These kinds of systems are essential steps towards the creation of a unified planning interface, especially when humans are involved in the loop. For a system aiming to provide the complete planning pipeline to a user, a key require-

ment for the XAIP community is the creation of systems to deliver explanations to users in an interactive and intuitive manner. Towards this goal, researchers have created systems using explanations for human-in-the-loop planning. For example, RADAR (Sengupta et al. 2017) and RADAR-X (Karthik et al. 2021) make use of contrastive explanations in addition to plan suggestions to develop decision-support systems for interactive explanatory dialogue with users. Another recent system (Eifler and Hoffmann 2020) discusses the design of an iterative planning interface that takes user preferences into account while helping them create plans via plan property dependencies. While these systems make use of interactive user interfaces, and the latter system uses a visualization to show plan execution, they all present explanations in text, and do not focus on how effectively the explanations are delivered. To the best of our knowledge, this paper is the first attempt to investigate to what extent visualizations are an effective medium for conveying explanations to users in an MRP setting.

## Preliminaries

### Classical Planning

A *classical planning* problem, typically represented in PDDL (Ghallab et al. 1998), is a tuple $\Pi = \langle D, I, G \rangle$, which consists of the domain $D = \langle F, A \rangle$ – where $F$ is a finite set of fluents representing the world states ($s \in F$) and $A$ a set of actions – and the initial and goal states $I, G \subseteq F$. An action $a$ is a tuple $\langle pre_a, eff_a^\pm \rangle$, where $pre_a$ are the preconditions of $a$ – conditions that must hold for the action to be applied; and $eff_a^\pm$ are the addition ($+$) and deletion ($-$) effects of $a$ – conditions that must hold after the action is applied. The solution to a planning problem $\Pi$ is a plan $\pi = \langle a_1, \ldots, a_n \rangle$ such that $\delta_\Pi(I, \pi) = G$, where $\delta_\Pi(\cdot)$ is the transition function of problem $\Pi$. The cost of a plan $\pi$ is given by $C(\pi, \Pi) = |\pi|$. Finally, a cost-minimal plan $\pi^* = \operatorname{argmin}_{\pi \in \{\pi' | \delta_\Pi(I, \pi') = G\}} C(\pi, \Pi)$ is called an optimal plan.

### Model Reconciliation Problem

A *model reconciliation problem* (MRP) (Chakraborti et al. 2017) is defined by the tuple $\Psi = \langle \Phi, \pi \rangle$, where $\Phi = \langle M^R, M_H^R \rangle$ is a tuple of the agent's model $M^R = \langle D^R, I^R, G^R \rangle$ and the agent's approximation of the human's model $M_H^R = \langle D_H^R, I_H^R, G_H^R \rangle$, and $\pi$ is the optimal plan in $M^R$. A solution to an MRP is an explanation $\epsilon$ such that when it is used to update the human's model $M_H^R$ to $\widehat{M}_H^{R,\epsilon}$, the plan $\pi$ is optimal in both the agent's model $M^R$ and the updated human model $\widehat{M}_H^{R,\epsilon}$. The goal is to find a cost-minimal explanation, where the cost of an explanation is defined as the length of the explanation.

In addition to adding information to the user's model, an explanation might also involve the removal of information from a user's model such that it is consistent with the agent's explanation (Vasileiou, Yeoh, and Son 2020). Therefore, our notion of explanation is defined as follows:

**Definition 1** (Explanation)**.** *Given an agent $M^R$, a user $M_H^R$, and an optimal plan $\pi$, assume that $\pi$ is only optimal*

---

[4]As there is a fast-growing amount of work on MRP and XAIP in general, we refer the reader to the survey by Chakraborti, Sreedharan, and Kambhampati (2020) for more information.

[5]We use both Planimation and Conductor as inspiration for the VizXP framework and discuss the details in a later section.

in $M^R$. Then, $\epsilon = \{\epsilon^+, \epsilon^-\}$ is an explanation from $M^R$ to $M_H^R$ for $\pi$ if $\pi$ is optimal in $\widehat{M}_H^{R,\epsilon} = (M_H^R \cup \epsilon^+) \setminus \epsilon^-$, where $\epsilon^+ \subseteq M^R$ and $\epsilon^- \subseteq M_H^R$,

As such, $\epsilon^+$ is the addition of information to the user's model and $\epsilon^-$ is the removal of information from the user's model.

## Taxonomy of Explanations

Most MRP algorithms look at explaining either optimal or valid plans to human users (Chakraborti et al. 2017; Vasileiou, Yeoh, and Son 2020). Towards that end, such explanations, using insights from social sciences (Miller 2019), are considered according to three main properties: *Social explanations* for modeling the expectations of the explainee; *selective explanations* for choosing the explanations among several competing hypotheses; and *contrastive explanations* for differentiating properties of two competing hypotheses. Among these properties, contrastive explanations have received a lot of attention (Hoffmann and Magazzeni 2019). However, all explanations share two common elements; They either express discrepancies between the domain-action models of the agent and the user (i.e., *domain-based* explanations) or involve differences in the initial and/or goal state assumptions of the planning problems of the agent and the user (i.e., *problem-based* explanations). Below, we formalize these two notions as characteristics of explanations stemming from MRP scenarios.

**Domain-based Explanations:** Assume an agent $M^R$, a user $M_H^R$, and a plan $\pi$ that is optimal in $M^R$ but not $M_H^R$. We say that an explanation from $M^R$ to $M_H^R$ for $\pi$ is a *domain-based* explanation, denoted by $\epsilon_d$, if all of its elements involve the action dynamics in $M^R$ and/or $M_H^R$. In other words, the elements of the explanation must involve addition (or removal) of actions, preconditions of actions, or effects of actions to (or from) $M_H^R$. More formally,

**Definition 2** (Domain-based Explanation). *Given an explanation $\epsilon_d$ from $M^R$ to $M_H^R$ for $\pi$, we say that $\epsilon_d = \{\epsilon_d^+, \epsilon_d^-\}$ is a domain-based explanation if $e^+ \subseteq A^R$ for all $e^+ \in \epsilon_d^+$ and $e^- \subseteq A_H^R$ for all $e^- \in \epsilon_d^-$, where $A^R$ and $A_H^R$ are the set of actions in $M^R$ and $M_H^R$, respectively.*

Note that we make the assumption that explanations involving the addition or removal of an entire action can be specified as a set of preconditions and/or effects accompanied by the name of the action.

**Problem-based Explanations:** Assume an agent $M^R$, a user $M_H^R$, and a plan $\pi$ that is optimal only in $M^R$ but not $M_H^R$. We say that an explanation is a *problem-based* explanation, denoted by $\epsilon_p$, if all of its elements involve the addition (or removal) of initial and/or goal states to (or from) $M_H^R$. More formally,

**Definition 3** (Problem-based Explanation). *Given an explanation $\epsilon_p$ from $M^R$ to $M_H^R$ for $\pi$, we say that $\epsilon_p = \{\epsilon_p^+, \epsilon_p^-\}$ is a problem-based explanation if $e^+ \subseteq I^R \cup G^R$ for all $e^+ \in \epsilon_p^+$ and $e^- \subseteq I_H^R \cup G_H^R$ for all $e^- \in \epsilon_p^-$, where $I^R \cup G^R$ and $I_H^R \cup G_H^R$ are the unions of the initial and goal states in $M^R$ and $M_H^R$, respectively.*

These categories make intuitive sense as any planner takes, as input, a domain file and a problem file, which fully specify the planning problem $\Pi$. We will utilize these two types of explanations in our visualization framework and posit that they are a suitable categorization for visualizing explanations for MRPs. We also note that these two kinds of explanations are not isolated, and some MRPs can have solutions that include both types of explanations. Further, the information provided in all explanations discussed above falls into one of the following two categories:

- **Action-space Information:** Given a planning problem $\Pi$, and an associated domain $D$ containing actions $A = \langle pre_A, eff_A^{\pm} \rangle$, the action-space information corresponds to information about the preconditions and effects for each action in $D$. Domain-based explanations will contain this kind of information.
- **State-space Information:** Given a planning problem $\Pi$, a plan $\pi$, and a sequence of states $S$ involved in the execution of $\pi$, the state-space information corresponds to information about the predicates in each state in $S$. Problem-based explanations, which address errors in the initial and goal state, contain this kind of information.

It is easy to see the parallels between the two kinds of explanations and the two kinds of information discussed above. As one may need to convey both types explanations when explaining a plan, an ideal system for presenting explanations should be able to convey both types of information to the users. In the next section, we discuss two existing visualization systems that present action-space and state-space information, and use those ideas to motivate the design of a framework capable of visualizing plans and their execution as well as presenting explanations, using both state-space and action-space information.

## Visualization Framework

Our goal was to create a set of guidelines that system designers can utilize when deploying a visualization system for presenting explanations to users. Borrowing elements from existing work in plan visualization, such a framework should be able to show all kinds of explanations discussed in the previous section. Given an explanation based on the user's plan and the agent's plan, it should support the visualization of the following information for the human user's model: (1) Plan length; (2) Wrong/missing initial/goal state; (3) Wrong/missing preconditions; (4) Wrong/missing effects; and (5) Wrong/missing actions.

As noted earlier, most MRP-based explanations are *contrastive* and, typically, involve a *foil* provided by the human user in terms of an alternative plan (Sreedharan, Srivastava, and Kambhampati 2018). In addition, the context of the user's own plan may help them in better understanding the agent's explanation. Hence, the user's plan provides an excellent window for presenting explanations, a fact that is useful for the visualization techniques proposed in the following sections.

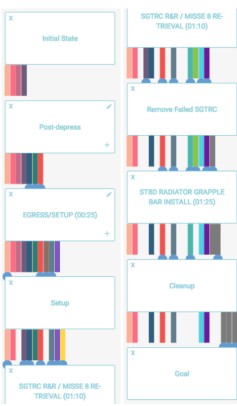

Figure 1: Illustration of Conductor (Bryce et al. 2017).

## Action-space Visualization: Fact Flows

"Fact routes" from Conductor (Bryce et al. 2017) provide an easy way to visually represent preconditions as "stations" on each action that need to be filled, and effects as routes originating from that action. Conductor, combined with Marshall (Bryce, Benton, and Boldt 2016), is aimed at helping users create domains and plans concurrently. Using fact routes to show the evolution of facts over time, it aims to use interactions with users to facilitate creating correct plans and domains. We found that Conductor's framework was limited by the fact that the length of the plan, as well as the number of predicates involved, increases the number of fact routes to the extent that it might overwhelm users (see Figure 1). This makes it unsuitable for all but the simplest of domains that contain few predicates and have short plans.

To remedy this, we introduce a simplification to Conductor. Instead of tracking all fact routes as individual columns, we visualize just the routes moving into and going out of each action as *fact flows*. Optionally, a user interaction like a click may show the history of the fact route for any particular action, thus retaining all relevant information for users who require it. This reduces clutter and allows us to present longer plans with domains that can contain larger number of predicates within a limited space. For example, consider a fact flow in(truck1, city2), and an action that does not use truck location as precondition; Conductor would show this fact flow before the action, while in our simplification, this unnecessary fact flow would be hidden.

In order to visualize explanations, we propose two methods: (1) *Highlight-based* and (2) *Port-based* methods. Using the former, the precondition/effect flows are highlighted based on whether they are unaffected (colored grey), wrong (colored red), required/missing (colored yellow), or required/present (colored green). The latter method employs "ports" for the preconditions and effect of each action, an extension of the "stations" used in Conductor. Ports can be colored based on whether they are unaffected (colored blue), wrong (colored red) or required (colored green), and fact flows can be either missing (not plugged in) or present (plugged in). Figure 2 shows an example of the same information conveyed using both methods.

One additional modification we make to Conductor's design is the introduction of the fact flows to the initial state. Instead of visualizing the entire initial state, we only show predicates that are affected by the explanation (e.g., problem-based explanation for the initial state).

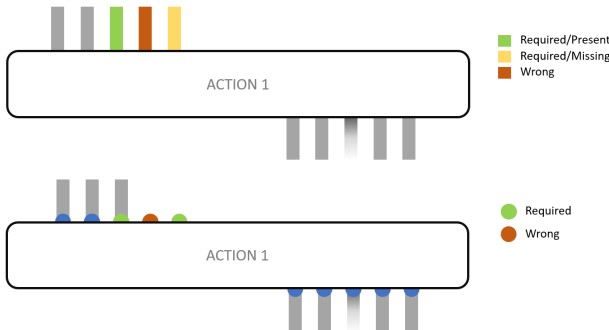

Figure 2: An action-space visualization example, with preconditions at the top of each action, and effects at the bottom. Deletion effects are represented as a flow fading out. Top: The highlight-based explanation visualization; Bottom: The port-based explanation visualization.

## State-space Visualization: Abstraction

While the action-space framework is sufficient to visualize all explanations, it fails to show information about the state of the world at certain times throughout the execution of the plan. Many planning domains contain features that can enable humans to think about them in terms of physical abstractions. Simple classical domains like BlocksWorld and Logistics naturally lend themselves to the physical space, presenting users the ability to keep track of the current state of the world by tracking their positions in their mental space. Moreover, planning visualization interfaces like Planimation (Chen et al. 2020) and WebPlanner (Magnaguagno et al. 2020) utilize state-space visualizations to assist in planning and display plan execution. Planimation, in particular, allows users to create visualizations for plans using an animation profile to specify how different elements are visualized.

Inspired by such systems, we propose an abstraction-based plan visualization which we extend to display explanations as well. We describe states and transitions between states using *containers* (objects in the world that can "contain" others), contents (objects that can be "contained" in others), and links (ways for contents to move between containers). We note that state-space visualizations like Planimation also fall within the framework described here.

**Abstract Space:** The positional relationships between various objects (e.g., On, In, etc.) and the motion of objects between *containers* form the basis of the state-space abstraction visualization. We present one hierarchy based approach for visualizing state-space information for planning domains that possess these kinds of relationships. Concretely, this approach requires the following properties:

- Domain objects are classified as either *containers*, *contents*, or both.
- Domain objects are either *movable* objects or *immovable* objects.
- All domain actions must move items between containers.

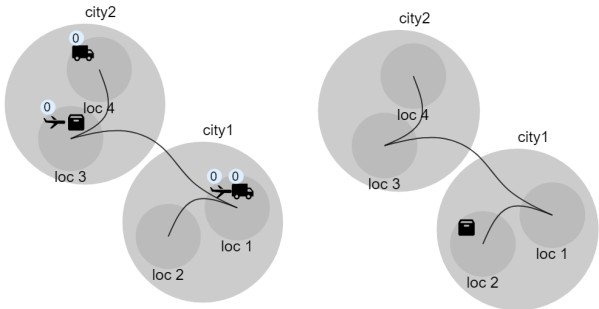

Figure 3: A state-space visualization example. Left: The initial state; Right: The goal state.

- Predicates must identify and fully specify the relationship between objects for any state.

Note that in some cases it might be necessary to introduce pseudo-predicates to allow for the last property. For example, in BlocksWorld, `onTable(a)` can be "reified" to `onTable(a, table)` with "table" being a dummy object created to represent the implicit table. This is only needed for the visualization, and need not change the planning process.

Any planning domain satisfying the above properties can be used to create a visual representation of the state of the world at any given step. We can visualize a network of containers connected by edges that movable contents or containers can traverse, with each edge-type represented by certain actions (e.g., in Logistics, the `move-airplane` action moves an airplane between two locations), with each action causing an object to move across one of these edges, with optional animations. This is a basic setup and may be specialized and modified for each domain. For example, Figure 3 shows the initial/goal states for a Logistics problem.

Within the state-space visualization, it is much easier to see "why" some positional relationships are not true. Simple preconditions like the requirement for different actions to have objects "in" certain locations are intuitively shown in the state if true, and effects of an action can be clearly seen with the motion of objects across these edges. This can help users during plan creation.

For presenting explanations within the state-space visualization, we employ the highlighting technique discussed in the action-space visualization. For each state in the execution of the plan, starting from the initial state, we display the current state with respect to the actions that are executed in the human's plan, using the agent's domain. Each object involved in a missing/wrong precondition or effect is shown similarly to the highlight-based approach in the action-space visualization.

## Integrated Action- and State-space Visualization

We now present *Visualizations for eXplainable Planning* (VizXP), a visualization framework that combines the action-space and state-space elements discussed previously. It can visualize plans and their execution as well as present explanations to human users, using both state-space and action-space information. The inclusion of the action-space information also conveniently presents a simple way for users to select and view different states. Highlights in the action-space visualization provide an overview of the steps where the users' plan went wrong, with the state-space visualization providing more detail about what exactly went wrong. In addition, VizXP also allows users to debug and correct their plans during the creation phase.

Finally, we note that depending on the application, an additional visualization might present the agent's correct plan alongside the human's plan, similar to contrastive explanation methods. This can then be used to display the 'required' information with the human's plan only visualizing the missing and wrong information. This is required for domain-based explanations that involve actions not in the user's plan.

## Evaluation Setup: User Study

We now discuss the setup for our evaluation, where we compared VizXP against a text-based benchmark, an approach commonly used by current state-of-the-art systems (Eifler and Hoffmann 2020; Karthik et al. 2021), through a user study conducted on the online crowdsourcing platform Prolific (Palan and Schitter 2018). The goal of the evaluation is to investigate to what degree MRP explanations presented by VizXP are effective and easily understood by humans compared to the text-based benchmark. Based on insights from other research communities, such as the multimedia learning principled described in the Introduction section, we hypothesize that *participants will perform significantly better with VizXP compared to the text-based baseline*.

As existing MRP solvers require that the explaining agent knows both its "correct" model and the "wrong" model of the human user receiving the explanation, we needed a mechanism to enforce this assumption. To do this, we used a simplified Logistics domain (McDermott 2000) as the "correct" model of the explaining agent, tweaked that model by removing some preconditions and changing the initial state, and assigned this tweaked "wrong" model to participating users. This assignment is done by describing the tweaked model to the users at the start of the study and asking them to create a plan for this wrong model. Then, users were provided MRP explanations and were asked to answer a series of questions as well as correct their plans based on those explanations. The users' answers to those questions as well as their ability to correct their plans reflect their understanding of the explanations provided.

**Domain and Problem:** Our choice of domain was the Logistics domain (McDermott 2000), which we simplified to make it less complex for people with no background in planning. Predicates `in-city`, `in`, and `at` were combined into one `in` predicate to avoid confusion. We renamed `airports` to `hubs` and changed the corresponding predicates to allow for some ambiguity to introduce errors in the domain. We created a simple problem with two cities containing two locations each. One location within each city is a hub. Figure 3 shows the initial and goal states for this problem. There are two airplanes and two trucks distributed across the locations, and one package that needs to be transported to the goal city. We considered two changes for the

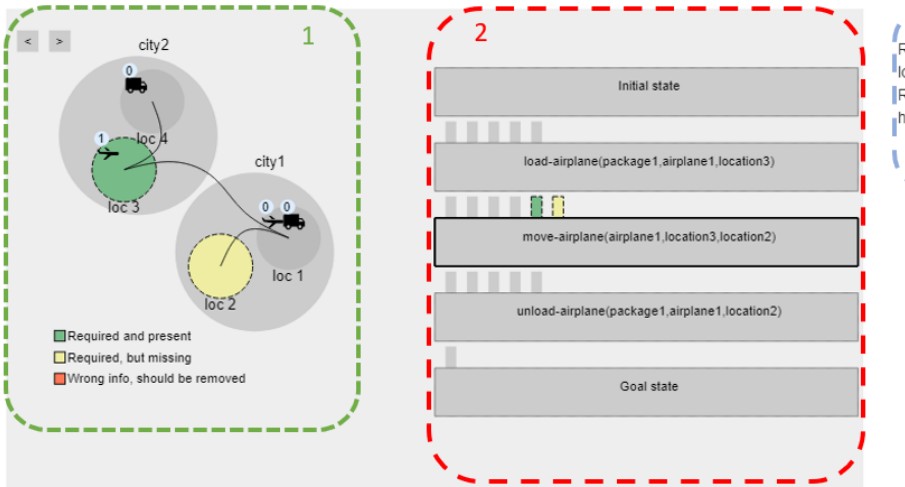

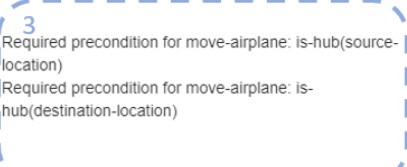

Figure 4: A view of the explanation visualization in the user study. (1) The state-space visualization; (2) The action-space visualization; (3) The text-based explanation.

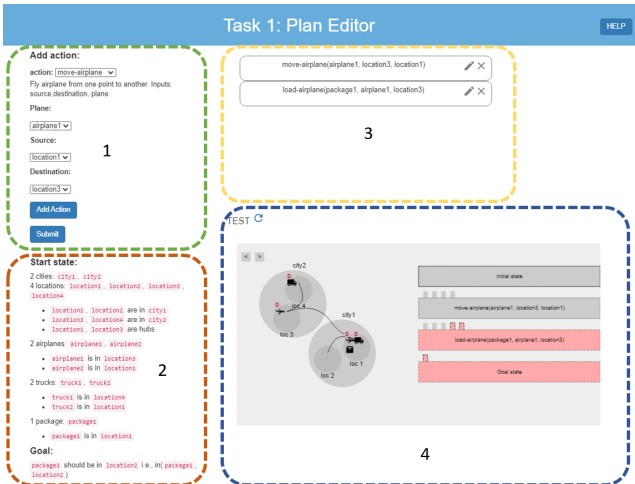

Figure 5: A view of the plan editor for the user study. (1) Action selection; (2) The initial and goal states; (3) User's current plan; (4) Test visualization showing validity of the plan.

"wrong" model of the user:

- **C1:** We modified the action `move-airplane` by removing its precondition that the source and destination location must be hubs. Therefore, a domain-based explanation is needed to correct this error.
- **C2:** We changed the initial location of the package, thereby requiring a problem-based explanation to correct this error.

**Prototype Implementation:** We used elements from VizXP to create a visualization system for the selected domain. For the state-space visualization, we used circles to mark cities and locations and icons for trucks, airplanes, and pack-

ages. There were two types of links: Transporting objects between locations (visible); and loading and unloading packages from trucks or airplanes (invisible). An alternate design could further separate these out by type, having different edges for the `move-airplane` action and for the `move-truck` action, but we chose to use only two kinds for the sake of simplicity. We used animations to show objects moving between containers. For the action-space visualization, we used a limited version of the system where only the flows into the current action are visualized instead of the entire fact route like in Conductor. Since none of the explanations would involve any change to the effects of actions, we decided to omit effect flows from the visualization. Figure 4 shows a view of the visualization presenting an explanation. We created the implementation to run on a browser, using Flask and Python as back-end, and D3.js and JavaScript for the front-end.

As VizXP is agnostic to the choice of algorithm to generate MRP explanations, we used one of the existing state-of-the-art solvers to generate the explantions. To display the explanations, we chose the highlight-based approach, with tooltips providing additional information.

**Study Design:** The study was designed to have two groups: The experimental group using VizXP and the control group using text. Each group was tested on two types of "wrong" models, modified using changes **C1** and **C2** (see "Domain and Problem" paragraph), each requiring a different type of explanation. Therefore, we have **four scenarios** in total, which we tested independently. We created two tasks for each user as follows:

- **Task 1:** For this task, participants were asked to create a plan based on the modified domain and problem information provided to them using a simple plan editing interface. This interface also allows users to "test" their

Table 1: User Study Results.

| | | Population Size | Correction Ratio | Correction Time (mean) | Comprehension Score |
|---|---|---|---|---|---|
| VizXP | all users | 86 | 0.698 | 203.35 | 5.402 |
| | computer science users | 29 | 0.793 | 199.84 | 5.400 |
| | domain-based explanations | 43 | 0.674 | 193.23 | 5.091 |
| | problem-based explanations | 43 | 0.721 | 213.74 | 5.720 |
| Text | all users | 83 | 0.627 | 251.05 | 4.759 |
| | computer science users | 41 | 0.585 | 246.98 | 4.340 |
| | domain-based explanations | 40 | 0.625 | 252.36 | 4.475 |
| | problem-based explanations | 43 | 0.628 | 249.84 | 5.023 |

plans, which will provide information about the errors in their plans due to their misunderstanding of the provided domain and problem information. Depending on the scenario, this interface might be either VizXP[6] (shown in Figure 5) or a sequence of steps with markers for incorrect actions. A participant succeeded in Task 1 if they created and submitted a valid plan given their domain and problem. Users that succeeded in Task 1 continue to Task 2, and users that failed in Task 1 were filtered out and ignored. This is important since MRP explanation-generation algorithms assume that the user's model is known.

- **Task 2:** For this task, we informed the participants that the initial domain and problem information provided to them contained errors and presented explanations for those errors using either VizXP or text based on the group of the participant. They were then asked a series of questions to evaluate their understanding of the explanation provided (**Task 2a**). Then, they were shown the plan editor again and asked to correct their plan, this time without the ability to "test" their plans for correctness (**Task 2b**). A participant succeeded in Task 2b if their corrected plan is valid in the agent's model.

To incentivize participants to provide answers to the best of their ability, we provided a bonus to participants who succeeded in Task 1 and an additional bonus to participants who also succeeded in Task 2b. Further, we also included two questions for attention checks in the study, where participants were asked to type a particular string or select a particular answer in a multiple choice question. Participants who wrongly answered both of these questions were filtered out of the study.

Each participant had the following interactions in the study: (1) They arrive at the webpage following the link from Prolific, where they enter their demographics and some information on their educational background. (2) To ensure that they have the background necessary to solve the tasks, they are given tutorials on classical planning, the logistics domain, and the plan editing interface. (3) Following the tutorials, they are asked to complete Task 1. (4) If they succeeded in Task 1, they are asked to complete Tasks 2a and 2b. (5) All participants, including those who failed Task 1, are

---

[6]Users in the experimental group are shown VizXP in Task 1 to ensure that they are familiar with the system before receiving an explanation using that interface to eliminate any learning effects.

then asked to provide feedback on the system's usability (Holzinger, Carrington, and Müller 2020) and are informed of their payments before being redirected back to Prolific.

**Participants:** We conducted the study with 200 participants (66 female, 132 male, 2 non-binary) with each of the four scenarios getting 50 random participants. Out of the 200 participants, only results from 169 participants were used as 30 failed Task 1 and one participant succeeded in Task 1 but wrongly answered the questions on attention checks.

**Measures:** To measure comprehension of explanations provided, we used the following measures:

- **Correction Ratio:** Proportion of users who succeeded in Task 1 who also succeeded in Task 2b.
- **Correction Time:** Time taken by users who succeeded in Task 2b in correcting their plan.
- **Comprehension Score:** Number of questions users answered correctly in Task 2a.

## Evaluation Results

We now discuss the results of our evaluations using the measures above to evaluate the performance of VizXP in aiding users understand explanations provided. For statistical significance, we used a $p$-value of 0.05 as a threshold.

Table 1 summarizes our results for four different groups of users who succeeded in Task 1: *all users*, the subgroup of users with a *computer science* (CS) background, the subgroup of users who were given the model with change **C1** and *domain-based explanations*, and the subgroup of users who were given the model with change **C2** and *problem-based explanations*. For each group of users, we report the population size of that group and our three measures. We now discuss the results for each of those measures:

- **Correction Ratio:** More users were able to accurately correct their plans with VizXP (= 69.8%) than with the text-based baseline (= 62.7%). However, the difference is not statistically significant ($\chi^2 = 0.6652$ and $p = 0.4147$ with two-proportion $z$-tests). Among the subgroup of users with a CS background, the difference is larger – 79.3% of users succeeded in correcting their plans with VizXP compared to 58.5% with the text-based baseline. This difference is *more* statistically significant ($\chi^2 = 2.4477$ and $p = 0.1177$). The likely reason is that a fraction of users without a CS background failed to sufficiently understand the planning problem and succeeded

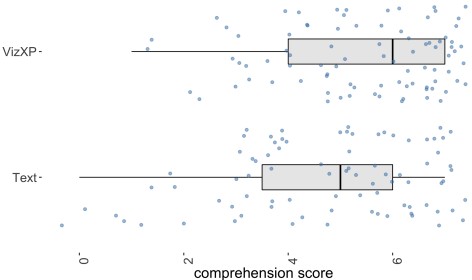

Figure 6: Comprehension Score Distribution for All Users.

in Task 1 due to the aid of the "test" functionality in the plan editing interface. And this fraction of users is similar across both VizXP and text-based baseline groups.

- **Correction Time:** In general, users were able to correct their plans faster with VizXP (average time of 203.35s) compared to with the text-based baseline (251.05s). However, the difference is not statistically significant ($\chi^2(1, N = 112) = 0.4901$, $p = 0.4839$, and $\epsilon^2 = 1$ with Kruskal-Wallis H non-parametric tests). This trend is also consistent for the subgroup of users with a CS background. We do not observe the improved performance of VizXP with this measure because the "test" functionality is absent in Task 2b. Thus, users who succeeded in Task 1 due to that functionality are not considered here.

- **Comprehension Score:** Similar to the previous two measures, users scored better on this measure with VizXP (= 5.402 out of 7 questions answered correctly on average) compared to with the text-based baseline (= 4.759). Unlike the other two measures, *this difference is statistically significant* ($\chi^2(1, N = 193) = 5.2252$, $p = 0.0222$, and $\epsilon^2 = 0.0371$ with Kruskal-Wallis H non-parametric tests). This difference and statistical significance is further amplified among the subgroup of users with a CS background. Figure 6 plots the distribution of comprehension scores for all users.

The trends above generally apply for the two subgroups who were given domain- and problem-based explanations also. However, there are not much noticeable differences between the two subgroups, indicating that VizXP (and the text-based baseline as well) perform equally well for both subgroups.

## Discussions

While the statistically significant results with the comprehension score measure are consistent with our expectations, we were surprised by the lack of statistical significance on the results with the correction ratio and correction time measures. We suspect the reason is that a non-trivial number of users succeeded in Task 1 despite not understanding the planning problem well due to the aid of the "test" functionality due to the following observations combined:

- The difference in the correction ratio is more statistically significant for the subgroup with a CS background, but the difference in the correction time is not more statistically significant for the same subgroup.
- All users who succeeded in Task 1 were included in the

correction ratio measure, but only users who succeeded in both Tasks 1 and 2b were included in the correction time measure.

Further, the statistical significance tests used are sensitive to the population sizes. Should the correction ratios remain unchanged for larger population sizes, then the differences between the users using VizXP and the users using the text-based baseline will also become more statistically significant. Therefore, we anticipate that the results for the correction ratio measure will be statistically significant with a larger user study and a better way of ensuring that users sufficiently understand the planning problem.

Additionally, we were surprised to find that 11 users answered at least 6 of the 7 comprehension questions correctly, implying that they understood the explanations well, but failed to accurately correct their plans. This observation implied that their error is due to typos and not misunderstanding of the explanations. This observation thus hints that the comprehension score measure, for which VizXP is statistically better than the text-based baseline, is more accurate at measuring how well users understand the explanations provided than the correction ratio measure.

Finally, we would also like to highlight that while user studies have been conducted in the XAIP literature, they are at a significantly smaller scale as they are meant to be feasibility studies only. For example, Eifler and Hoffmann (2020) and Chakraborti et al. (2019) conducted user studies with only 6 and 39 participants, respectively. Therefore, this paper spearheads the important need for larger-scale user studies that are necessary for measuring the efficacy of explanations with human users, laying critical foundations for interactive two-way dialogues with users in future XAIP systems.

We note some limitations of this work as well. In the user study, we require users to create full alternate plans, but in many contrastive explanation systems, users are also able to use partial foils. It is possible to envision a system designed using VizXP that can use partial foils by splitting the plan at the point of interest and comparing the partial plans, but further work will be required to test that ability and its applicability to real systems.

Additionally, the scope of the container based visualization needs to be better defined. For example, it is not trivial to fit object properties like color or capacity (e.g. fuellevel in NoMystery) into the container framework. However, not all information needs to be captured in the abstraction. With simple augmentations to denote properties (like a blip with current fuel), even such properties can be described in the state space visualization. However, that is domain-specific, and thus will need to be done on a case-by-case basis.

## Conclusions

In this paper, we proposed VizXP, a visualization framework for visualizing MRP explanations. Through a combination of state-space and action-space visualizations, we showed how one can visualize both domain-based and problem-based explanations. Through a comprehensive user study, we evaluated the performance of VizXP and found that users, on average, understood explanations better when us-

ing VizXP than when using a text-based baseline, which is commonly used by existing state-of-the-art systems. Further, the improvement of VizXP over the baseline is even more pronounced in users with a computer science background, indicating its usefulness for experienced users. In conclusion, this paper makes the important contribution of improving the medium by which explanations are conveyed to users, orthogonal to most existing work focusing on advancing the state of the art in generating explanations, and laying the necessary foundations for successful deployment of XAIP systems in the real world.

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
