# OpenReview forum: "VizXP: A Visualization Framework for Conveying Explanations to Users in Model Reconciliation Problems"
_icaps-conference.org/ICAPS/2021/Workshop/XAIP — XAIP 2021_

### Official Review · AnonReviewer1 · 2021-07-04
**An interesting visualization framework that provides explanations in Model Reconciliation Problems**

**Rating:** 7
**Confidence:** 5

**Review:**

## Paper Summary

The paper proposes a visualization framework to convey explanations in Model Reconciliation Problems. The visualization framework is called VizXP.
Based on existing Planning visualization interfaces in the literature, VizXP combines of state-space and action-space visualizations to provide explanations in Model Reconciliation Problems. An evaluation with several participants (with different backgrounds) shows that VizXP provides understandable explanations when compared to a text-based baseline.

## Review

Please, consider the following minor points and comments.

### Minor Points and Comments:

1. You could try to place Figure 1 on page 4, where it is referenced in the text. Also, Figure 3 is only referenced on page 5, and you could try to place it closer to where it's referenced.

2. Does the explanation visualization interface have tooltips to show the value of the preconditions and effects? It would be good if the explanation visualization provides tooltips to see the value preconditions and effects, so it will help the user to actually know what is missing or wrong in the action description.

3. Figure 5 shows the Plan Editor. The editor is a bit inconsistent with some names and labels. For instance, it displays "Start State" and "Initial State", "Goal" and "Goal State", "location" and "loc". Such inconsistencies may be confusing for users that are not familiar with the Planning notation. Please, try to fix these inconsistencies.

4. The user study results show that the participants understood explanations better when using VizXP. However, although VizXP has better results than the text-based baseline for all used measures, the results of the text-based baseline are competitive with the results of VizXP. Don't you think that the explanation visualization interface should be improved to have even better results? I mean, the interface could be improved not only with respect to the way the interface conveys the explanations, but also regarding the usability of the interface to correct plans and modify domains and problems.

## Final Remarks

To conclude, I think that the visualization framework proposed in the paper is a valuable contribution to the XAIP community, so I recommend the acceptance of the paper.
Please, try to address the minor points and comments that I pointed out above in the next revision/version of the paper.

---

### Official Review · AnonReviewer2 · 2021-07-06
**The paper is relevant to the workshop, however I believe the study conducted for the evalation needs to be improved.**

**Rating:** 6
**Confidence:** 3

**Review:**

Summary --

The paper is motivated to create an effective method of communicating explanations. They compare two modes -- (1) text-based and (2) visual explanation using text and images. The authors have created a system to generate visual explanation (called VizXP), which they use to explain using text and images. VizXP generates images to explain using Conductor (Bryce et al. (2017)) to provide contrastive explanations using the model reconciliation process to present either the domain knowledge or the state information. The effectiveness of VizXP is evaluated through a user study where they perform two tasks. In the first task, they create a plan using an incorrect model, and in the second task, they provide answers after getting the explanation and fix the plan (constructed in task 1) in the correct model.

The paper seems relevant to XAIP as the mode of explanation is essential and very little research has been done on the subject. However, I believe the user study in it's current form was not correctly conducted, and the results are not generally applicable. Please see my review comments below.

Review Comments --

(1) I would like to read more about the user study, especially how the domain was explained to the users for task 1 (maybe I just missed it). This method is critical to the user study; ideally, they would want the understanding of the domain to be similar for every user before starting the tasks. What if they explained the domain using texts or through images that would bias the user to certain kinds of explanations.

(2) The study follows the pre-test (construct a plan), treatment followed by post-test methodology (correct the plan), which is the right way to evaluate the treatment. However, users who did not create the right plan were removed (per the incorrect model). It ensures that explanations are given to users who are good enough to understand them. At this point, to the best of my knowledge, I can safely say that the study is biased. Please remember that the same methodology has to be followed for every user, and the results should be reported for all of them. ``Knit-picking'' the users for a specific use-case should not be done for any user-study as the results are not acceptable in such a case. You usually pick users, for example, based on demographics when the study is for a specific demographic. In this case, the study evaluates explanations for people with the critical ability to construct a plan.

(3) FWIW, the model reconciliation process, as described in the seminal paper (Chakraborti et al. (2017)) is when the human-in-the-loop and the agent have different models and capabilities. In this paper, the authors assume that the robot and humans have the model for the same environment. Thus reconciliation happens between the robot's model and the robot's understanding of the human model. This assumption is not mentioned in the paper and might be helpful to understand. This kind of reconciliation is usually common in kind of teaching process.

(4) There have been some other user studies that are missing the related works section -- (Zahedi et al. (2019)), (Grover et al. (2020)). There is a lot of literature in the Intelligent Tutoring System community for providing feedback to students, that could be relevant to the authors such as (only to get you started) -- (Poulos & Mahony (2008)).

(5) I would also like to read more about migrating state-based explanations to different domains. For example, the authors had to use descriptive images such as a plane or a truck that are domain-specific. Maybe a paragraph on what would be needed to create an interface for other domains can be useful.

References --

[1] Ann Poulos & Mary Jane Mahony (2008) Effectiveness of feedback: the students’ perspective, Assessment & Evaluation in Higher Education, 33:2, 143-154, DOI: 10.1080/02602930601127869

[2] Z. Zahedi, A. Olmo, T. Chakraborti, S. Sreedharan and S. Kambhampati, "Towards Understanding User Preferences for Explanation Types in Model Reconciliation," 2019 14th ACM/IEEE International Conference on Human-Robot Interaction (HRI), 2019, pp. 648-649, doi: 10.1109/HRI.2019.8673097.

[3] Grover, S., Sengupta, S., Chakraborti, T., Mishra, A. P., & Kambhampati, S. (2020). RADAR: automated task planning for proactive decision support. Human–Computer Interaction, 35(5-6), 387-412.

---

### Meta-Review · Area_Chairs · 2021-07-07

**Recommendation:** Accept
**Confidence:** 4

**Metareview:**

Thank you very much for your submission.

The paper addresses the question of how an explanation can be communicated to the user. This is essential and very little research has been done on this topic.

The main discussion points are the setup and evaluation of the user study:

1. Introducing bias into the user study by filtering users based on their ability to construct plans.
2. Results of the text-based baseline are competitive with the results of VizXP. What are possible improvements to the interface?

We hope the reviews will be helpful. Please consider the comments for the camera-ready version. We look forward to your presentation!

---

### Decision · Program_Chairs · 2021-07-08

Accept